# Bullying and Self-Concept, Factors Affecting the Mental Health of School Adolescents

**DOI:** 10.3390/healthcare11152214

**Published:** 2023-08-06

**Authors:** Carmen Galán-Arroyo, Santiago Gómez-Paniagua, Nicolás Contreras-Barraza, José Carmelo Adsuar, Pedro R. Olivares, Jorge Rojo-Ramos

**Affiliations:** 1Physical and Health Literacy and Health-Related Quality of Life (PHYQoL), Faculty of Sport Science, University of Extremadura, 10003 Cáceres, Spain; mamengalana@unex.es; 2Sport, Health & Exercise Research Unit (SHERU), Castelo Branco Polytechnic Institute, School of Education, Department Sport and Well-Being, 6000-266 Castelo Branco, Portugal; 3BioẼrgon Research Group, Faculty of Sport Sciences, University of Extremadura, 10003 Cáceres, Spain; 4Facultad de Economía y Negocios, Universidad Andrés Bello, Viña del Mar 2531015, Chile; nicolas.contreras@unab.cl; 5Promoting a Healthy Society Research Group (PHeSO), Faculty of Sport Sciences, University of Extremadura, 10003 Cáceres, Spain; jadssal@unex.es; 6Faculty of Education, Psychology and Sport Sciences, University of Huelva, 21007 Huelva, Spain; pedro.olivares@ddi.uhu.es; 7Facultad de Educación, Universidad Autónoma de Chile, Talca 3480094, Chile; 8Physical Activity for Education, Performance and Health Research Group, Faculty of Sport Sciences, University of Extremadura, 10003 Cáceres, Spain

**Keywords:** bullying, self-concept, physical education, adolescents, associations

## Abstract

Bullying is an aggressive and repetitive behavior, where one person or several people physically, socially, or emotionally harm a vulnerable person and provokean imbalance of power in a school setting. Several factors such as age, sex, school performance, psychological factors, and ethnicity have been associated with bullying and more are being sought. Thus, the objectives of this study were as follows: (1) analyze the differences in bullying (victimization and aggression) and self-concept (academic, social, emotional, family, and physical) with respect to sex, school location, and educational level among Spanish adolescents; (2) explore the associations of bullying and self-concept with these sociodemographic dimensions. A cross-sectional study was designed with a total of 1155 participants (between 12 and 18 years old); there were 48.8% boys and 51.2% girls, where 75.9% studied compulsory secondary education (CSE) and 24.1% Baccalaureate, and 31.9% were students from rural schools and 68.1% were from urban schools. Medium and inverse correlations were shown between victimization and self-concept at the general level, for both sexes, both types of school, and both educational stages. For the aggression dimension, the correlations with self-concept were inverse at the general level (low), in girls (low), in rural students (medium), and in compulsory secondary education students (medium). For academic self-concept and family self-concept, the associations were medium and inverse with bullying in all variables. For emotional self-concept the correlation with bullying was direct and medium in all variables; in physical self-concept, the correlations with bullying were inverse in almost all variables except in boys. Self-concept may be a protective factor for bullying and interventions should aim at adolescents building a positive multidimensional self-concept that prevents and protects them from bullying either as aggressor or victim.

## 1. Introduction

The origins of research on bullying date back to the 1970s [1]. From that moment on, a prolific field of research began to grow rapidly in recent years [2]. Just as scientific production in this area continues to grow, so does its prevalence, associated risk factors, consequences, and forms of prevention and intervention. The scientific production in this regard comes to place bullying as a multicausal phenomenon that occurs in the school environment and constitutes a relevant public health problem [3]. It can be defined as an aggressive and repetitive behavior which is exercised by a person or group of people in the school environment with the intention of harming at physical, social, or emotional level another vulnerable person and cause in imbalance of power. It can take many forms, such as verbal or physical intimidation, social isolation, exclusion, defamation, and cyberbullying [4,5,6]. According to Anatalis et al. [7], the prevalence in Spain of students reporting bullying victimization is around 30%. In addition, 2.6% reported bullying perpetration and 15% reported being bullies/victims in primary education. This is also detailed in another study, where the prevalence of victimization ranges between 3% and 33% in adolescents aged 11 to 15 years, and an average rate of perpetrators of harassment of 20.6%, with a range between 9% and 54% [8].

The recent review by Cuesta et al. [9] highlights factors that have been associated with bullying such as age, sex, ethnicity, psychological factors, sexual activity, sexual/physical abuse, family factors, socioeconomic status, stressful events, and school performance, among others [10]. Regarding sex, scientific literature has pointed out that male stereotypes, still existing in developed societies, influence school-age youth negatively. Thus, they show behaviors related to violence and bullying, mainly since the most common behaviors of boys involve both the use of physical force and insulting or threatening a peer [11]. However, other research indicates that girls with more feminine attributes are more likely to experience victimization behaviors, as their actions tend to be more about exclusion or rejection by social groups, or about spreading rumors about their peers [12]. Similarly, the environment in which the school is located seems to be of vital importance when it comes to recording bullying behaviors and proposing social and educational strategies, either to prevent or combat it. Generally, the existing evidence has shown that abusive behaviors were more recurrent in urban areas compared to rural areas [13], although there are more current studies that indicate that bullying has been gradually spreading to these rural environments [14]. However, some experts point out that these changes in trends may be produced by the different classifications given to the environments internationally, as well as the different sociocultural variables affecting the populations studied [15]. In addition, the grade to which the students belong has been identified as a determinant factor in the expression of bullying-related behaviors [16], reaching its maximum expression during early adolescence and decreasing as adulthood approaches [17].

In this sense, the association between the five dimensions of self-concept (social, emotional, academic/work, family, and physical) and bullying-related behaviors has only recently begun to be explored by the scientific community [18]. Self-concept can be defined as the set of beliefs that the individual has of their own person at a given time, where the positive or negative feelings they have about themselves and the experiences lived in their social and cultural environment help to build their personality and develop emotionally and socially [19]. Self-concept, therefore, can be a relevant factor, especially in bullying victims, as they tend to have a negative view of themselves and their situation, and consequently a worse self-concept [10]. In addition, it can become a vicious circle in which students with a low self-concept will be further victimized, which will further damage their self-esteem [20] and could lead to more serious situations such as suicidal ideation and self-aggressive behavior [9,18]. In this regard, several studies have exposed among their results that victims of bullying tend to have a more negative self-concept compared to their peers who do not suffer bullying [21,22,23]. More specifically, Houbre et al. [24] showed that bullying victims possessed worse self-concepts about their social competence, physical appearance, global self-esteem, and self-efficacy than their non-bullied peers. In line with the above, recent studies on the analysis of research on bullying have detailed that there is a certain tendency to study bullying with aspects such as acceptance, resilience, self-esteem, or social support [25]. On the other hand, the relationship between self-concept and the abuser’s aggressive behaviors has also received attention from researchers. Normally, the bully’s self-concept is reinforced by the bullying behaviors, being socially accepted by his peers, and seeing himself as a superior person in comparison to his victims and, therefore, motivated to maintain his behaviors [26]. However, abusive behaviors and self-concept are also negatively associated, as research has shown that abusers have a poorer family and academic self-concept, affecting their relationship with their family and school environment [27].

Therefore, this research aims (1) to explore bullying (victimization and aggression) and self-concept (in relation to academic, social, emotional, family, and physical domains) as a function of sex, school location, and educational stage in Spanish adolescents. In addition, the aim is (2) to explore the possible correlations between bullying and self-concept, specifically analyzing the associations between its subdomains as a function of the aforementioned variables.

It is hypothesized that (1) both bullying and self-concept of Spanish adolescents will be affected by their sociodemographic variables: with respect to bullying, girls will show more victimization and boys more aggressiveness; students in urban schools might be more aggressive than in rural schools; and the older the school age, the more aggressive behavior. With respect to self-concept, there will be higher academic and emotional self-concept in girls and higher social, family, and physical self-concept in boys; students in rural schools will have higher self-concept than in urban schools; and the higher the level of education, the better the self-concept. It is also believed that (2) there will be an inverse correlation between bullying-related behaviors and self-concept, with the magnitude varying according to socio-demographic variables. In this way, the aim is to identify the current state of both bullying behaviors and students’ self-concept, in order to develop multidimensional educational and/or social strategies to improve them both in the short and long term.

## 2. Materials and Methods

### 2.1. Participants

The inclusion criteria for participants established in this study were as follows: (1) submit an informed consent form, where parents or guardians authorize the student to participate in the research; and (2) be a student of physical education at a public or private secondary school (from 12 to 18 years old) in Extremadura, an Autonomous Community of Spain.

A non-probability sampling method based on convenience sampling was used to determine the sample size [28]. The sample consisted of 1155 students, 48.8% of whom were boys and 51.2% girls, which is an equal proportion in terms of the sex of the participants. For the educational stage, the proportion was more disproportionate, where 75.9% were students in Compulsory Secondary Education (CSE) (12 to 16 years old) and 24.1% were Baccalaureate students (17 and 18 years old). In relation to the location of the school, 31.9% belonged to rural schools and 68.1% to urban schools, classifying as urban schools those located in towns with more than 20,000 inhabitants and as rural schools those located in towns with fewer than 20,000 inhabitants. These categories were established following the criteria of the Diputación Provincial de Cáceres (https://www.dip-caceres.es/, accessed on 1 June 2023). Finally, 75.2% studied in public schools and 24.8% in private schools, and all this sociodemographic information can be found in Table 1.

### 2.2. Procedure

It was possible to establish which institutions offer physical education courses for Secondary Education through Baccalaureate (from 12 to 18 years of age) by gaining access to the Department of Education and Employment’s database. The physical education teachers employed at these schools were emailed to inquire about the possibility of organizing a researcher visit so they could give the questionnaire to the pupils who had their parents’ informed consent. They were asked to use the same channel to reply to the questions as well. The communication included a description of the study’s goals, the parents’ informed consent, and the instrument models that had been utilized. If the teachers agreed to work together, they were required to reply to the email by arranging for a researcher to come to the school and, after obtaining parental permission, interview the pupils about bullying. The students were given first access to the questionnaire through a tablet, and each item was thoroughly explained to them one at a time so they would not be confused when responding. After collecting all of the questionnaires, the researchers processed, cleaned, and anonymized the data in order to prepare it for a second researcher’s subsequent blind analysis.

The Biosafety and Bioethics Committee of the University of Extremadura in Spain (Registration Code 71/2022) has approved a protocol that follows the principles of the Declaration of Helsinki.

### 2.3. Instruments

#### 2.3.1. Sociodemographic Data

In order to acquire sociodemographic data about the participating sample, a preliminary questionnaire was first developed that addressed questions about the students’ sex, educational stage, school environment, and school type.

#### 2.3.2. Bullying-Related Behaviors

With the objective to analyze bullying-related behaviors, the European Bullying Intervention Project Questionnaire (EBIPQ) in Spanish was used [29]. This scale was previously validated in the school population, and it has 14 items spread across 2 main dimensions, 7 of which (Dimension 1) indicate victimization-related characteristics and 7 of which (Dimension 2) relate to aggressiveness. The actions, which pertain to both dimensions, include beating, insulting, threatening, stealing, swearing, excluding, or spreading rumors. Each question has a Likert scale format with a score between 0 and 4, where 0 means “Never” and 4 means “Always,” with a time range of the two months prior. Each dimension obtains a different score, ranging from 0 (few victimization/aggression behaviors) to 28 (recurrent victimization/aggression behaviors). For the calculation of the total score, the overall scores of both dimensions are added together since victimization and aggression behaviors are not mutually exclusive.

#### 2.3.3. Self-Concept

Finally, the AF-5 scale [18], which was also validated in the study population, consisting of 30 total items covering 5 dimensions—academic self-concept, social self-concept, emotional self-concept, family self-concept, and physical self-concept—was used to assess self-concept. The scale is from 1 to 5, with 1 denoting “strongly disagree” and 5 denoting “strongly agree.” The authors reported that the AF-5 scale delivers indices higher than 0.71 in relation to the psychometric properties of the instrument in each of the five domains. Each dimension has its own score ranging from 5 (poor self-concept) to 25 (optimal self-concept), where the final score of the scale is the average value of the 5 dimensions that belong to it. It was demonstrated that all scale components measure the same construct (self-concept), yielding a value of 0.78 when taken as a whole.

### 2.4. Statistical Analysis

First, the Kolmogorov–Smirnov test was used to check whether the data of the continuous variables followed a normal distribution. This was not the case, so we proceeded to use nonparametric statistical tests. The Mann–Whitney U-test was used to analyze possible differences in the scores of the dimensions of the EBIPQ questionnaire and the AF-5 scale as a function of sex, school location, and educational stage. Spearman’s Rho test was used to explore the correlations between the scores of each of the dimensions of the questionnaires. The thresholds proposed by Mondragón-Barrera [30] were taken into account to interpret the following correlation coefficients: from 0.01 to 0.10 (low correlation), from 0.11 to 0.50 (medium correlation), from 0.51 to 0.75 (considerable correlation), from 0.76 to 0.90 (very high correlation), and from 0.91 to 1.00 (perfect correlation). In addition, in order to analyze the reliability of each of the instruments, Cronbach’s alpha, McDonald’s omega, and composite reliability were used. The values cited by Nunnally Berstein [31] were used as a reference to interpret the values resulting from the reliability test: <0.70 (low), 0.71 to 0.90 (satisfactory), and >0.91 (excellent). The SPSS statistical software version 23 for MAC (IBM SPSS, Chicago, IL, USA) was used to process the data collected.

## 3. Results

Firstly, Table 2 shows how all the reliability values obtained for the different dimensions of the scales of both the EBIPQ and the AF-5 can be considered satisfactory.

Table 3 shows both the descriptive statistics and the differences obtained in the two dimensions of the EBIPQ according to sex, school environment, and the educational stage to which the students belong. In general, sex showed significant differences in both dimensions, with girls showing higher scores in the victimization dimension of the EBIPQ, while boys students obtained higher scores in the aggression dimension. Likewise, the school location only showed significant differences in the aggression behavior dimension, with students from urban schools having higher values. Similarly, educational stage only showed differences in the second dimension, with Baccalaureate students showing higher ratings compared to their CSE peers.

Likewise, Table 4 contains the descriptive statistics of the results obtained in AF-5 by the participants. All the dimensions of this scale showed significant differences according to the sex of the students, so that boys perceived a better social, family, and physical self-concept. However, the girls expressed better ratings of their academic and emotional self-concept. On the other hand, the school location did not seem to influence the students’ self-concepts, except in the family dimension where students from rural schools had better scores than their urban peers, giving rise to significant differences. Finally, the educational stage only highlighted differences with respect to the emotional dimension, pointing to high school students as those with a better self-concept.

As for the correlations existing between the EBIPQ dimensions and the AF-5 score (Table 5), both were characterized as inverse and significant; however, the dimension referring to victimization exhibited a medium association while that of aggression was defined as low. In addition, sex exhibited mean, inverse, and significant correlations for EBIPQ victimization and self-concept, exhibiting a stronger relationship in girls. Regarding the aggression dimension, only the female sex highlighted significant differences, showing low and inverse associations in both sexes with the self-concept scale. On the other hand, the location of the school showed significant inverse mean relationships between the victimization dimension and self-concept, with a slightly higher coefficient in students from urban schools. However, aggression behaviors are only significantly associated with self-concept in students in rural schools, inversely and averagely. Finally, the section on victimization generated inverse, mean, and significant relationships with self-concept in both educational stages, with the Baccalaureate stage exhibiting the highest coefficient. In contrast, aggression behaviors had significant, mean, and inverse associations with self-concept in CSE students.

Finally, the possible correlations between the values corresponding to the dimensions of the AF-5 scale and the final EBIPQ scores were explored (Table 6). In general, all dimensions of the AF-5 were significant when associated with the EBIPQ. The academic, family, and physical self-concepts showed inverse and average relationships, while the social dimension, although also inverse, was characterized as low. Emotional self-concept revealed a medium and direct correlation. With respect to sex, all the dimensions of the AF-5 showed significant relationships with the EBIPQ, except for social and physical self-concept in male students. In this sense, all the associations in the female population have higher values compared to the male population, most of them being inverse except for emotional self-concept. Next, the location of the school yielded different results, with only one non-significant correlation, that between social self-concept and EBIPQ in both types of schools. The urban students found higher values than their rural peers in the academic and emotional self-concepts. In contrast, family and physical self-concepts are higher in students belonging to rural schools. Finally, the results of the educational stage yielded significant results in all dimensions except social self-concept, most of them being mean and inverse, since in emotional self-concept they were direct. Higher values were found in CSE students for the academic, emotional, and family dimensions; however, the physical self-concept showed a higher coefficient in Baccalaureate students.

## 4. Discussion

First, significant differences were observed for the two dimensions of bullying, with girls showing higher scores on the victimization dimension and boys showing higher scores on the abuse dimension. These results are supported by the findings of other studies, which have shown that girls are more involved in the role of victim and boys in the role of bully [5,32,33]. This could be due to a certain social expectation about the stereotypical participation of boys and girls in bullying situations [6]. In this sense, Navarro et al. in their study showed how stereotypical characteristics of masculinity are more related to bullying perpetration and violence [11], and Silva et al. showed how feminine traits of girls are more related to victimization [34]. These differences could also be explained by the fact that boys mostly use physical violence, threats, or insults in bullying and girls are associated with relational behaviors such as being ignored, exclusion, or spreading rumors [12,35]. For school location, students studying in urban schools showed significantly higher values on the aggression dimension than those studying in rural schools. In this sense, the study by De Frutos et al. [13] supports this fact, showing that bullying is more common in schools located in an urban environment than those located in rural environments. However, more recent research shows contradictory results, indicating a higher prevalence in rural areas [14,36]. These differences with respect to other studies could be explained by how the various studies define the concept of rural or urban and by the cultural influence of the country or region where the study was conducted [15]. In relation to the educational stage, Baccalaureate students showed significantly higher scores in the aggression dimension than CSE students. The results are certainly contradictory to previous research, which shows that the results are higher for the role of the bully between 11 and 15 years of age, or the periods ranging from 1st to 4th grade of CSE [16,37], with the highest involvement occurring in middle adolescence and then decreasing in later years [17]. This could be due to the type of bullying and the specific behaviors employed as adolescents grow older, evolving from more aggressive and violent bullying to more indirect and subtle forms [5,38].

In this study, significant differences were also observed in the five dimensions of self-concept according to sex, with better academic and emotional self-concepts in girls and better physical, family, and social self-concepts in boys. The results of this study are in line with the scientific evidence, with boys showing a better physical self-concept than girls [39,40,41]. Biological aspects could influence this self-concept, with boys achieving better marks in most physical skills and possessing greater muscle density than girls [42]. The physiological variations that occur during puberty could also have a certain relationship, occurring earlier in girls than in boys [43] and the latter trying to hide these changes because they consider them less attractive, with negative repercussions on their physical self-concept [44]. For academic self-concept, the results of other studies also corroborate the findings of this study, with girls having a better academic self-concept than boys [40,41,45]. This academic self-concept could be influenced by the greater dedication and self-demanding nature of girls in the academic field [46], as well as their better study habits, which usually lead them to obtain better grades than boys [47]. In this manner, it has been shown that girls associate academic success with effort and boys with their available skills [48]. However, in emotional self-concept, the results shown in this study are contrary to later research where a better emotional self-concept is observed in boys [41,45,49]. In relation to this, some research shows that boys seem to manage their negative emotions better [50] and other studies show that women have better emotional skills than men [51]; therefore, there is some uncertainty regarding emotional self-concept. In the social and family self-concepts, the results of other studies are also contrary to ours, showing that girls have a better social and family self-concept or, directly, no significant differences are observed between the sexes [40,41,45,49]. In the research by Herrera et al., they explain that the higher family and social self-concepts of girls could be associated with the fact that they show better social skills such as empathy and social responsibility, while boys show better emotional skills such as stress tolerance and impulse control, which could influence their better emotional self-concept [52]. The treatment received by parents and the close environment could also have an impact in this sense [52]. For the educational stage, no significant differences were found for self-concept in most dimensions between CSE and Baccalaureate individuals, except in emotional self-concept, where a better emotional self-concept was observed in Baccalaureate students. The scores between both stages are very similar, which could be due to the small age difference between the two stages. This is the opposite case to that observed when comparing self-concept between secondary and primary education, since in secondary school there are psychological and emotional alterations that build the self-concept and personality of the individual [53]. Finally, the location of the school does not seem to influence the students’ self-concept, except in the family dimension, where students from rural schools showed a better self-concept than those from urban schools. In this sense, communication and a good family environment favor the development of self-esteem and self-concept of adolescents in both rural and urban areas [54]. However, this better family self-concept of rural students could be due to the fact that life in these areas is less complex, since rural adolescents are subject to less pressure and stress due to their family environment, in addition to the fact that their family relationships could be closer [55].

In general, the victimization dimension (medium association) and the aggression dimension (low association) of the EBIPQ were inversely and significantly associated with the total score of the AF-5 self-concept scale, which means that the higher the self-concept, the lower the aggression and victimization, with victimization having a stronger association. In both boys and girls, victimization had an inverse, medium, and significant association with self-concept, with the coefficient being higher in girls; however, in the aggression dimension, only in girls was it inversely and significantly associated with self-concept, though this was a small association. As for school location, students from rural schools showed an inverse and average association between both dimensions of bullying and self-concept, and students from urban schools only showed an inverse and average association for victimization. The same occurred with the educational stage, with CSE students showing an inverse and average association between both dimensions of bullying and self-concept, while Baccalaureate students only showed it for victimization, where the coefficient was higher than those of CSE. In the case of the association between victimization and self-concept, several studies corroborate these results, stating that students with a low self-concept will be more victimized; this fact will have even more negative repercussions on their self-esteem, and may even lead, in extreme cases, to suicidal ideation and self-injurious behaviors [10,56]. Furthermore, high levels of self-concept would have a protective effect on bullying victimization, since a sense of competence and positive self-efficacy about oneself could contribute to the fight against depressive symptoms that could result from bullying victimization [10,57,58]. Regarding the association between self-concept and the aggression dimension, this study showed a low and inverse association; however, there is some uncertainty and few studies that analyze the multidimensional self-concept in traditional bullies. Some research contradicts our results, as the study by Olthof et al. shows how bullies are socially accepted and are motivated to denigrate their victims, presenting a higher social self-concept [26]. The same happens with the physical self-concept in the study by Taylor et al. where bullies had a positive self-image of themselves, seeing themselves as superior to their victims [20]. On the other hand, other studies support our findings, stating that bullies present a negative family self-concept generated by a family climate lacking affection and little communication, and a lower academic self-concept [59], possibly due to lower academic performance [27].

Finally, the association between the five dimensions of the AF-5 scale and the final EBIPQ scores was studied. For academic self-concept, an inverse and average association was shown at the general level, for both sexes, for both types of school, and for both educational stages, presenting very similar coefficients. Other studies support this as we have cited above in the case of aggressors who showed a more negative academic self-concept [60], which could be due to poor academic performance [27]. A lower academic self-concept also produces greater victimization [59], where negative experiences, low grades, and negative perceptions at school together with a poor school climate could be associated with bullying and have a negative impact on academic self-concept [60,61]. For the social self-concept, the association with the EBIPQ total score was low-to-inverse at the general level, and it was only associated with girls in this same way. Furthermore, in the case of bullying victims, several studies show how victims have a lower social self-concept than students who are not involved in bullying [59,62], which may be influenced by the prolonged rejection received by their peers that generates feelings of social incompetence and perception of loneliness and triggers greater isolation [63]. However, as mentioned above, in the case of the aggressors the scientific evidence contradicts our results, showing a better social self-concept that could be favored by the social acceptance of their environment [26]. In the emotional self-concept, a direct and average association was obtained with the EBIPQ at the general level and for all the variables of sex, location of the school, and educational stage. These results are striking, since a better emotional self-concept is related to a greater involvement in bullying. Most research, both in the cases of victims and aggressors, exposes a lower emotional intelligence that could influence the construction of a lower emotional self-concept [48,64]. In the case of aggressors, this is shown by not being aware of the negative consequences of their actions and possessing little empathy [48], and in the case of victims by less emotional clarity and not being able to regulate and interrupt negative emotions [64]. The best coefficients were obtained for family self-concept, showing an inverse and average association at the general level and with all variables. Several studies support these data, especially in offenders who present a lower family self-concept [59,65], possibly characterized by a poor affective bond between parents and children [66] and by the frequent presence of family conflicts [67]. For physical self-concept, the association with EBIPQ was medium and inverse at the general level, for girls, for students in rural schools, and for both educational stages, with this association being low for students in urban schools. The study by Benítez-Sillero et al. follows the line of our study, showing a negative association between physical self-concept and victimization, also exposing physical activity as a protective aspect against bullying, showing a strong association between physical activity and self-concept [68].

### 4.1. Practical Implications

This study has shown the association between adolescent self-concept and bullying, either as a victim or as an aggressor. It is suggested that policies and programs be implemented and developed based on these findings, where adolescents can apply tools and techniques to work on their self-concept in the different dimensions that compose it. Along these lines, it has been proven that self-concept and its different dimensions are made up of different social, mental, emotional, and physical factors, where educational schools could exert some influence, adopting a competent teaching–learning process based on the interests and motivations of the students, as well as families that provide support, positive perceptions, and protection for the adolescents. In addition, interventions should pay special attention to emotional knowledge and regulation skills, helping victims to recognize what they feel and become socially, mentally, and physically resilient. Such interventions would also help bullies to control their emotions and impulses where the use of any type of violence is prohibited, decreasing arousal generated by frustration or distress.

### 4.2. Limitations and Futures Lines of Research

This research has certain limitations, like any other. For example, caution should be exercised when interpreting the results of this study since it is a cross-sectional and correlational study. Also, the sample was only composed of students from Extremadura, a Spanish autonomous community. Similarly, the role of the bystander was not taken into account in this study because the questionnaire used did not contemplate it. In addition, only quantitative methods were used, although qualitative methodology could yield important information on multidimensional processes, such as the development of self-concept and the existence of bullying-related behaviors.

As possibilities for future project, it is suggested to conduct a longitudinal study with measurements at different times in order to establish the observed relationships over time. Additionally, it would be interesting to apply this study in other Spanish regions or in other countries, analyzing whether there are sociocultural differences. In the same way, other research could study the role of the bystander and its possible association with self-concept. The fear of defending the victim for becoming a new target of the bully could influence the results to some extent [69]. Finally, these questionnaires could be applied to the entire school environment (teachers, parents, administrators, etc.) in order to understand the different perspectives of educational agents, identifying the current state and developing strategies based on this.

## 5. Conclusions

Significant differences were obtained for both dimensions of bullying according to sex, with girls showing more victimization and boys more aggression. Students from urban schools showed more aggressive behavior than their rural peers, and Baccalaureate students showed more aggression than CSE students. A higher academic and emotional self-concept was observed in girls and a higher social, family, and physical self-concept was observed in boys. School location and educational stage did not seem to influence self-concept, since only a higher family self-concept was observed in rural students with respect to urban students and a higher emotional self-concept in Baccalaureate students with respect to CSE students. Victimization was inversely associated with self-concept, at the general level, in both sexes, in both types of school, and in both educational stages. However, the aggression dimension was only associated with self-concept at the general level, in girls, in rural students, and in CSE students. Academic and family self-concept showed inverse associations with bullying at the general level, in both sexes, in both types of school, and in both educational stages. The same was true for physical self-concept, with the exception of boys, where the correlation was not significant. Emotional self-concept showed direct associations with bullying at the general level and for all variables. Social self-concept only showed correlation with bullying overall and in girls. These findings show how self-concept can be a protective factor for bullying. Therefore, interventions should be carried out from different areas to cover the different dimensions of self-concept and for adolescents to build a positive multidimensional self-concept that protects them from being involved in bullying situations either as aggressor or victim.

## Figures and Tables

**Table 1 healthcare-11-02214-t001:** Sociodemographic characteristics of the participants (*n* = 1155).

	Categories	*n*	%
Sex	Boys	564	48.8
	Girls	591	51.2
Educational Stage	CSE	877	75.9
	Baccalaureate	278	24.1
School Location	Rural	368	31.9
	Urban	787	68.1
School type	Public	869	75.2
	Private	286	24.8

*n*: number, %: percentage, CSE: Compulsory Secondary Education.

**Table 2 healthcare-11-02214-t002:** Reliability indices for the different dimensions belonging to the scales used.

Dimensions	Cronbach’s Alpha	McDonald’s Omega	Composite Reliability
EBIPQ			
Victimization	0.847	0.850	0.850
Aggression	0.851	0.856	0.856
AF-5			
Academic	0.878	0.884	0.884
Social	0.758	0.789	0.789
Emotional	0.801	0.800	0.800
Family	0.879	0.879	0.886
Physical	0.763	0.766	0.766

**Table 3 healthcare-11-02214-t003:** Descriptive results of the EBIPQ according to sex, school location, and educational stage.

	Sex			School Location			Educational Stage		
Item	Girls Me (IQR)	Boys Me (IQR)	*p* (U)	Rural Me (IQR)	Urban Me (IQR)	*p* (U)	CSE Me (IQR)	Baccalaureate Me (IQR)	*p* (U)
1. Victimization	1.86 (0.7)	1.71 (0.8)	<0.01 **	1.86 (0.9)	1.86 (0.9)	0.28	1.86 (0.9)	1.86 (0.9)	0.80
			(172,325)			(155,547)			(120,705)
2. Aggression	1.43 (0.6)	1.43 (0.7)	<0.01 **	1.29 (0.6)	1.43 (0.7)	<0.01 **	1.43 (0.6)	1.50 (0.7)	<0.01 **
			(170,160)			(147,278)			(104,567)

Me = median value; IQR = interquartile range. Differences are significant at ** *p* < 0.01. Each score is obtained is based on a Likert scale (0–4): 0 is “Never” and 4 “Always”.

**Table 4 healthcare-11-02214-t004:** Descriptive results of the AF-5 according to sex, school location, and educational stage.

	Sex			School Location			Educational Stage		
Dimensions	Girls Me (IQR)	Boys Me (IQR)	*p* (U)	Rural Me (IQR)	Urban Me (IQR)	*p* (U)	CSE Me (IQR)	Baccalaureate Me (IQR)	*p* (U)
1. Academic	3.83 (1.0)	3.66 (1.2)	<0.01 **	3.66 (1.2)	3.66 (1.0)	0.26	3.66 (1.0)	3.66 (1.2)	0.59
			(164,023)			(157,911)			(119,316)
2. Social	3.83 (1.0)	4.00 (0.8)	<0.01 **	3.83 (1.0)	3.83 (1.0)	0.92	3.83 (1.0)	3.83 (0.8)	0.16
			(155,380)			(158,255)			(115,060)
3. Emotional	3.00 (1.2)	2.33 (0.8)	<0.01 **	2.66 (1.0)	2.66 (1.2)	0.19	2.66 (1.0)	2.83 (1.2)	<0.01 **
			(114,336)			(151,260)			(108,394)
4. Family	4.50 (1.0)	4.66 (0.8)	<0.01 **	4.66 (1.0)	4.66 (1.0)	0.02 *	4.66 (1.0)	4.50 (1.0)	0.08
			(168,919)			(150,092)			(113,639)
5. Physical	3.33 (1.0)	3.83 (1.0)	<0.01 **	3.66 (1.1)	3.50 (1.2)	0.56	3.50 (1.0)	3.50 (1.2)	0.44
			(122,177)			(157,567)			(118,141)

Me = median value; IQR = interquartile range. Differences are significant at ** *p* < 0.01; * *p* < 0.05. Each score obtained is based on a Likert scale (1–5): 1 is “Strongly disagree” and 5 “Strongly agree”.

**Table 5 healthcare-11-02214-t005:** Correlations between EBIPQ dimensions and AF-5 score, according to sex, school location, and educational stage of the student body.

EBIPQ Dimensions	AF-5 *ρ*	Sex		School Location		Educational Stage	
		Girls Me (IQR)	Boys Me (IQR)	Rural Me (IQR)	Urban Me (IQR)	CSE Me (IQR)	Baccalaureate Me (IQR)
1. Victimization	−0.18 **	−0.23 **	−0.12 **	−0.17 **	−0.18 **	−0.16 **	−0.22 **
2. Aggression	−0.08 **	−0.09 *	−0.07	−0.18 **	−0.03	−0.11 **	−0.01

Me = median value; IQR = interquartile range. Differences are significant at ** *p* < 0.01; * *p* < 0.05.

**Table 6 healthcare-11-02214-t006:** Correlations between AF-5 dimensions and EBIPQ score, according to sex, school location, and educational stage of the student body.

Dimensions	EBIPQ *ρ*	Sex		School Location		Educational Stage	
		Girls Me (IQR)	Boys Me (IQR)	Rural Me (IQR)	Urban Me (IQR)	CSE Me (IQR)	Baccalaureate Me (IQR)
1. Academic	−0.24 **	−0.25 **	−0.20 **	−0.21 **	−0.25 **	−0.25 **	−0.18 **
2. Social	−0.07 *	−0.09 *	−0.04	−0.05	−0.08	−0.07	−0.07
3. Emotional	0.28 **	0.31 **	0.27 **	0.22 **	0.30 **	0.29 **	0.22 **
4. Family	−0.37 **	−0.45 **	−0.29 **	−0.41 **	−0.35 **	−0.39 **	−0.30 **
5. Physical	−0.11 **	−0.13 **	−0.04	−0.13 **	−0.09 *	−0.11 **	−0.12 *

Me = median value; IQR = interquartile range. Differences are significant at ** *p* < 0.01; * *p* < 0.05.

## Data Availability

The datasets are available through the corresponding author upon reasonable request.

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
