# Peer review of "Bullying and Self-Concept, Factors Affecting the Mental Health of School Adolescents"

_healthcare, 2023, doi:10.3390/healthcare11152214_

Round 1

Reviewer 1 Report

The theoretical framework, the methodology, the results, and their discussion are clearly presented. The authors reflect on the practical implications and analyze the limitations of the study. However, it is advisable to review the following aspects:

(i) The definition of bullying given in lines 21 and 22 of the abstract should be revised. As indicated in the theoretical framework, the imbalance of power between the victim and the aggressor associated with the phenomenon of bullying should be emphasized;

(ii) To clarify in the manuscript the terminology adopted (gender or sex), since these concepts are used in an undifferentiated way;

(iii) In the citations presented in the text the year should be added;

(iv) In Table 1, to clarify the meaning of the acronym CSE- Compulsory Secondary Education;

(v) In Table 2, delete from the table legend "* p < 0.05 ". Also, the asterisk is missing in “<0.01” when presenting the "School Location" data;

(vi) In line 187 the authors allude to "FY-5" but in the manuscript they refer to the instrument as "AF-5".

Author Response

The following document shows the modifications made.

Reviewer 2 Report

Thank you for the opportunity to review the manuscript entitled, "Bullying and self-concept, factors affecting the mental health of school adolescents”. I believe this study investigated a topic relevant to the readers of “Healthcare”. 

Bullying is acknowledged around the world as a serious problem in educational settings during childhood and adolescence. Due to its high prevalence, bullying is an important topic of study. The victims are usually the target of hostile attacks without any incitement, systematically suffering the situation with a great degree of emotional intensity that can lead to severe states of anxiety or depression and social isolation.

This study aims to explore the possible correlations between bullying and self-concept.

The study involved a sample of 1155 students. In general, this paper is well written and follows well accepted standards of academic writing. However, the manuscript does not meet my expectations for publication in the journal. The manuscript lacks clear introduction and sophisticated and/or ambitious a statistical analysis.

Introduction:                                                                      

The introduction does not analyze in detail the variables under study. The introduction does not provide sufficient background. It lacks a robust conceptual framework. The discussion seems to contain information (studies) that would have been more appropriate in the Introduction.

Instruments:

It is necessary to use other alternative reliability indices, such as Composite Reliability (CR) and McDonald's Omega which are calculated through factorial loads and are measured more accurate reliability.

The instruments must always display two important qualities: reliability and validity. It is good practice to perform a confirmatory factor analysis.  Is need to assess the validity of the constructs of the scales (Confirmatory Factor Analysis: Relative Chi-Square, P; IFI; GFI; AGFI; CFI; RMSEA…)

Statistical analysis:

The study lacks Sophisticated and/or ambitious a statistical analysis. (Spearman's Rho correlation coefficient)

Thank you.

Author Response

(The authors gave the same response as above.)

Reviewer 3 Report

General appreciation

Bullying is a public health issue with detrimental effects at both the micro and macro levels. Therefore, further exploration of the psychological processes associated with bullying is extremely important. Additionally, self-concept is a structural dimension of psychological functioning with significant implications for socioemotional adjustment.

The manuscript has a clear structure and content, and while some changes are needed to improve its quality, it meets the general criteria for publication. Please find below some suggestions regarding the changes that I believe are necessary to make the manuscript clearer and more robust.

Abstract

The distinction between the first and second objectives is not clear. It would be relevant to include the age span of the participants. It is not usual to present the r and p values in the abstract, but rather only provide a description of the observed correlations.

Introduction

The structure of this section is clear, and the arguments are presented in a concise and objective manner. The authors describe evidence supporting gender differences regarding bullying. However, the results of research on the association of bullying with school location and education level are not documented. The authors' hypotheses are lacking. It is important to more clearly identify the gap that the study aims to fill in order to highlight its relevance.

Materials and Methods

In the participants' description, it is unclear whether the authors established any criteria regarding the students' age span. Additionally, what about the exclusion criteria? For example, were cognitive or reading/writing difficulties considered as exclusion criteria? It is necessary to clarify whether the research team ensured that all the students were able to autonomously complete the questionnaires.

Including subsections for each instrument or dimension assessed (i.e., sociodemographic information, bullying behaviors, and self-concept) would improve the organization of this section.

Information on the procedures required to compute the subscale scores and total scores of the EBIPQ and AF5 is missing.

Regarding the statistical analysis, it may not be necessary to include information on the analysis performed to assess the internal consistency of the instruments in this section. The authors only need to report the observed Cronbach's alpha values for the scales of each instrument in the description of the instruments and include the citations that theoretically support their interpretation.

Along with these issues, I have an ethical concern regarding data collection, as it is not clear whether the informed consent form included the possibility of benefiting from professional help, such as psychological support. For instance, if students have reported being victims of verbal or physical aggression by their peers, it would be crucial to offer them the opportunity to receive specific support.

Results

Statistical symbols must always be italicized. In Table 3, the U values should be included. Results concerning the internal consistency of the instruments (cf., lines 240-244) are typically presented in the Materials and Methods section at the end of the description of each instrument.

Discussion

In the practical implications’ subsection, I suggest using the expression "emotional knowledge" instead of "emotional understanding" as a core dimension of emotional competence, in line with Susanne Denham's proposal.

In the "Limitations" subsection, the second part of the first sentence (cf., lines 400-401) may be deleted, as referring to causal relationships does not make sense when studying multidimensional and complex phenomena such as bullying and self-concept. Furthermore, exploring additional limitations of the study and providing suggestions for future research will make this section more robust. For example, it could be mentioned as a limitation that only quantitative methodology (i.e., questionnaires) was used. Another limitation is that only students' perceptions were considered, and a multi-informant approach would have been beneficial.

Overall, addressing these issues and making the suggested revisions will enhance the clarity and quality of the manuscript, bringing it in line with the criteria necessary for publication.

A moderate gramatical revision is needed.

Author Response

(The authors gave the same response as above.)

Round 2

Reviewer 2 Report

Thank you for the opportunity to review again the manuscript entitled, "Bullying and self-concept, factors affecting the mental health of school adolescents”.

The authors took into account my comments and they proceeded with the necessary revisions. I thank the authors their time and effort. My opinion is that in the current form the manuscript could be published.

Author Response

REVIEWER 2

Comments and Suggestions for Authors

Thank you for the opportunity to review again the manuscript entitled, "Bullying and self-concept, factors affecting the mental health of school adolescents”.

The authors took into account my comments and they proceeded with the necessary revisions. I thank the authors their time and effort. My opinion is that in the current form the manuscript could be published.

 Author's reply: Thank you very much for your time and input. His contribution was decisive in improving the manuscript.

Reviewer 3 Report

The authors substantially improved the manuscript both theoretically and methodologically. However, to ensure that the manuscript meets all the criteria needed for publication, I recommend the authors to further clarify the study's objectives and hypotheses, particularly in the abstract and introduction. Additionally, the description of reliability indices needs attention. By addressing these remaining concerns, the manuscript will be publishable.

Abstract

The authors have incorporated the necessary changes, but the study objectives still require clarification. It is essential that the authors explicitly state their intention to: i) analyze/compare the differences in bullying (victimization and aggression) and self-concept (academic, social, emotional, family, and physical) with respect to sex, school location, and educational level among Spanish adolescents; ii) explore the associations of bullying and self-concept with these sociodemographic dimensions.

Introduction

The authors have provided additional evidence to support the association of bullying with school location and educational level. I recommend using the term "sex" consistently (cf., line 67). Instead of "academic bibliography," consider using "existing evidence" or a similar expression. Following the suggestions for the abstract, the objectives and hypotheses need correction to ensure consistency. Since the authors have performed comparative and correlational analyses, the predictions must be formulated accordingly. The first hypothesis should focus on the expected differences concerning sex, school location, and educational level. In the second hypothesis, the authors should indicate whether they expect to observe a positive or negative association of bullying behaviors and self-concept dimensions with the referred sociodemographic dimensions. The sentence concerning the practical implications of the study may be rephrased and included after the hypothesis (cf., lines 126-128).

Method

All the previously raised concerns have been properly addressed.

Results

The reliability indices of the instruments should be presented before the results of the comparative and correlational analysis, as it is standard practice to report the reliability of the instruments used to measure variables before presenting the results.

Discussion

All the identified issues have been corrected by the authors.

A grammatical revision will be useful.

Author Response

REVIEWER 3

Comments and Suggestions for Authors

The authors substantially improved the manuscript both theoretically and methodologically. However, to ensure that the manuscript meets all the criteria needed for publication, I recommend the authors to further clarify the study's objectives and hypotheses, particularly in the abstract and introduction. Additionally, the description of reliability indices needs attention. By addressing these remaining concerns, the manuscript will be publishable.

 Author's reply: Dear reviewer, thank you very much for your dedication, we have adhered to all your suggestions.

Abstract

The authors have incorporated the necessary changes, but the study objectives still require clarification. It is essential that the authors explicitly state their intention to: i) analyze/compare the differences in bullying (victimization and aggression) and self-concept (academic, social, emotional, family, and physical) with respect to sex, school location, and educational level among Spanish adolescents; ii) explore the associations of bullying and self-concept with these sociodemographic dimensions.

 Author's reply: Thanks, we have added to the abstract the specific objectives you recommended.

Introduction

The authors have provided additional evidence to support the association of bullying with school location and educational level. I recommend using the term "sex" consistently (cf., line 67). Instead of "academic bibliography," consider using "existing evidence" or a similar expression. Following the suggestions for the abstract, the objectives and hypotheses need correction to ensure consistency. Since the authors have performed comparative and correlational analyses, the predictions must be formulated accordingly. The first hypothesis should focus on the expected differences concerning sex, school location, and educational level. In the second hypothesis, the authors should indicate whether they expect to observe a positive or negative association of bullying behaviors and self-concept dimensions with the referred sociodemographic dimensions. The sentence concerning the practical implications of the study may be rephrased and included after the hypothesis (cf., lines 126-128).

 Author's reply: Thanks, We have modified the introduction with all your proposals.

Method

All the previously raised concerns have been properly addressed.

Author's reply: Thank you very much.

Results

The reliability indices of the instruments should be presented before the results of the comparative and correlational analysis, as it is standard practice to report the reliability of the instruments used to measure variables before presenting the results.

Author's reply: Perfect, thanks, it has already been modified as you suggested.

Discussion

All the identified issues have been corrected by the authors.

Author's reply: Thank you very much.
